# Removal of Chromium(VI) by Chitosan Beads Modified with Sodium Dodecyl Sulfate (SDS)

**Xiaoyu Du [1], Chihiro Kishima [1], Haixin Zhang [1], Naoto Miyamoto [2] and Naoki Kano [2,*]**

[1] Graduate School of Science and Technology, Niigata University, 8050 Ikarashi 2-Nocho, Nishi-ku, Niigata 950-2181, Japan; F17K502A@mail.cc.niigata-u.ac.jp (X.D.); c_kishima@nokkluber.co.jp (C.K.); F19B137H@mail.cc.niigata-u.ac.jp (H.Z.)

[2] Department of Chemistry and Chemical Engineering, Faculty of Engineering, Niigata University, 8050 Ikarashi 2-Nocho, Nishi-ku, Niigata city, Niigata 950-2181, Japan; nmiyamoto@eng.niigata-u.ac.jp

* Correspondence: kano@eng.niigata-u.ac.jp; Tel.: +81-025-262-7218

**Abstract:** In this study, chitosan beads modified with sodium dodecyl sulfate (SDS) were successfully synthesized and employed for the removal of chromium(VI) (Cr(VI)). The adsorption performance of the adsorbent (SDS-chitosan beads) was examined by batch experiments. The partition coefficient (PC) as well as the adsorption capacity were evaluated to assess the true performance of the adsorbent in this work. The adsorbent (SDS-chitosan beads) showed a maximum Cr(VI) adsorption capacity of 3.23 mg·g$^{-1}$ and PC of 9.5 mg·g$^{-1}$·mM$^{-1}$ for Cr(VI). The prepared adsorbent was characterized by different techniques such as scanning electron microscopy-energy dispersive X-ray spectroscopy (SEM-EDS), X-ray photoelectron spectroscopy (XPS) and Fourier transform-infrared spectroscopy (FT-IR). We used inductively coupled plasma mass spectrometry (ICP-MS) for the determination of Cr(VI) in solution. The experimental data could be well-fitted by pseudo-second-order kinetic and Langmuir isotherm models. The thermodynamic studies indicated that the adsorption process was favorable under the higher temperature condition. The SDS-modified chitosan beads synthesized in this work represent a promising adsorbent for removing Cr(VI).

**Keywords:** chitosan beads; sodium dodecyl sulfate (SDS); hexavalent chromium ions; adsorption capacity; partition coefficient

## 1. Introduction

Toxic metal contamination in aquatic environments has attracted tremendous attention due to the rapidly increasing number of manufacturing industries. Heavy metals are of major environmental concern because they are non-biodegradable and cannot be decomposed or metabolized [1]. Several metals cause serious health and environment problems, and chromium (Cr) compounds are one of the most toxic contaminants in wastewater due to their high solubility and toxicity, and free transferability [2]. Cr has been widely applied in many industrial activities based on its excellent properties, for example, such as in electroplating, leather tanning, nuclear power plants, and textile industries [3,4]. Moreover, it can be used for anodizing, corrosion control, and chemical manufacturing [5–7]. Chromium usually exists in two stable oxidation states: trivalent Cr(III) and hexavalent Cr(VI) in a natural environment, as its other oxidation states are not stable in aerated aqueous media [8]. In a natural aqueous environment, Cr(VI) may exist in the form of $CrO_4^{2-}$ or $HCrO_4^-$, whereas Cr(III) is inclined to form $[Cr(H_2O)_6]^{3+}$, $Cr(H_2O)_5(OH)^{2+}$, $Cr(H_2O)_4(OH)_2^+$, or Cr(III) organic complexes. It is well-known that Cr(III) is an essential material for organisms, whereas Cr(VI) is more toxic, carcinogenic, and mutagenic [8,9]. Thus, the development of a recovery method for this metal (particularly Cr(VI)) is significant from the standpoint of environmental protection. Many physical and



chemical technologies, such as ion exchange, precipitation, ultrafiltration, reverse osmosis, and electro dialysis, have been reported and used for removing heavy metals [10]. Nevertheless, these procedures have some disadvantages, such as a high consumption of reagents and energy, low selectivity, high operational costs and difficult further treatment due to the generation of toxic sludge [11]. Adsorption is considered to be an efficient method for removing metallic ions in aqueous solutions [12,13]; in particular, biological adsorption (biosorption) is one of the most environmentally friendly, economically favorable, low cost, recyclable, and technically easy methods [14,15].

Among the many biosorbents available, chitosan can be an outstanding biosorbent for metals for the reason that its amine (-NH$_2$) and hydroxyl (-OH) groups may serve as coordination sites to form complexes with various heavy metal ions [16]. Chitosan has been proven to be very efficient as a biosorbent for the recovery of several toxic metals such as mercury (Hg), uranium (U), molybdenum (Mo), vanadium (V), and platinum (Pt) [17–19], and its full chemical name is called (1,4)-2-amino-2-deoxy-β-D-glucose. It can be employed as an environmentally friendly adsorbent due to it being economical and the fact that it does not result in secondary pollution. Chitosan can be produced by the alkaline deacetylation of chitin, which originates from the most abundant biopolymer—cellulose. Chitosan is a polymer that can be acquired from the shells of seafood, such as prawns, crabs, and lobsters [20]. The biopolymer is characterized by a high percentage of nitrogen, and exists in the form of amine groups, free amino groups, and hydroxyl groups, which are responsible for metal ion binding through chelation mechanisms [21].

The uses of chitosan in the removal of various pollutants have been adequately reviewed [22]. However, chitosan has some defects, such as notable swelling in aqueous media and nonporous structures, resulting in a very low surface area [23]. Therefore, many types of chemical modification can be undertaken to produce chitosan derivatives for improving the removal efficiency of the heavy metal [24]. For example, various chemical or physical modifications can be adopted for increasing the number of exposed active sites [25,26]. Moreover, silicon dioxide can be employed to offset the defects of chitosan because it has many properties, such as a rigid configuration, porosity, and a high surface area. In the case of silicon dioxide, modified silicon dioxide produced through the graft between silanol groups and ligands has been developed [27–29]. We have synthesized a hybrid membrane of carboxymethyl chitosan and silicon dioxide as adsorbents for the removal of Cr(VI) in our previous work [30]. Furthermore, we have carried out an adsorption experiment of chromate ions onto cross-linked chitosan using epichlorohydrin (EP) and glutaraldehyde (GA) as cross-linked agents [31].

In this work, we evaluated the adsorption of chitosan modified with sodium dodecyl sulfate (SDS) as part of the adsorption study of Cr using modified chitosan. It has been reported that SDS-modified chitosan beads can be used to remove cationic dyes. [32]. The metal ion strength and the presence of key functional groups on the polymer chain allow its adsorption on surfaces [33–35]. The aggregation of particles through a bridging structure can be described as a two-step pathway: (1) initial chain adsorption and bridging, followed by (2) floc maturation/reconfiguration. Before the interparticle connection occurs, the chain of SDS must be adsorbed on a chitosan surface [36]. Furthermore, chitosan modified with SDS has recently been used for the removal of heavy metals, such as cadmium [37,38]. However, the use of SDS-modified chitosan as an adsorbent of Cr, with varying initial concentrations of SDS for optimizing the adsorbent, has rarely been evaluated. The objective of the present research was to investigate the efficiency of SDS-modified chitosan beads as a sorbent for Cr(VI) for more practical uses in the future, and to reveal the adsorption mechanism. After the characterization of SDS-chitosan by scanning electron microscopy-energy dispersive X-ray spectroscopy (SEM-EDS), X-ray photoelectron spectroscopy (XPS) and Fourier transform-infrared spectroscopy (FT-IR), batch experiments using SDS-modified chitosan beads were conducted to optimize the parameters, in order to obtain the maximum removal of Cr(VI). The effects of various parameters, such as the solution pH, contact time, adsorbent dose, and initial concentration, were examined.

## 2. Materials and Methods

Chemical reagents, including chitosan and sodium dodecyl sulfate (SDS; M.W.: 288.372 g/mol), were purchased from Tokyo Chemical Industry Co., Inc. (Tokyo, Japan). Acetic acid, NaOH, HNO$_3$, NaSO$_4$, ethylenediaminetetraacetic acid disodium salt dihydrate, and toluene were obtained from Kanto Chemical Industry Co., Inc. All reagents used were of analytical grade. During the whole working process, water (>18.2 MΩ) treated by the ultrapure water system (RFU 424TA, Advantech Aquarius) was employed. In order to prepare the Cr standard solution for the calibration curve, the standard solution (Kanto Chemical Co., Inc., 1000 mg· L$^{-1}$ K$_2$CrO$_7$ solution) was diluted and used.

### 2.1. Synthesis of the Adsorbent

Chitosan powder with a molecular weight (50–190 kDa) and degree of deacetylation (>80%) was used in this work. The viscosity of chitosan is 20 to 100mPa·s (in 0.5% acetic acid, 20 °C; degree Celsius) after drying. At first, 1.5 g of chitosan was placed in acetic acid solution (2.0%), and the solution was mixed for 24 h. The chitosan gel was prepared by dropping the above chitosan solution into 200 mL of 0.20 mol·L$^{-1}$ NaOH. The obtained gel was rinsed with ultrapure water until attaining a pH of 7 after stirring for 24 h. Subsequently, 200 tablets of chitosan gel beads were placed in 100 mL SDS solution (including the fixed concentration of SDS), and then, left for five days. Therefore, chitosan gel beads modified with SDS were obtained. Finally, the SDS-modified chitosan beads were dried at 60 °C overnight for their use as an adsorbent. The procedure employed for the synthesis of SDS-chitosan beads is shown in Figure 1. It is considered that the prepared adsorbent has a bilayer of SDS over the surface of pure chitosan beads. This bilayer can have a higher ion capturing capacity [37].

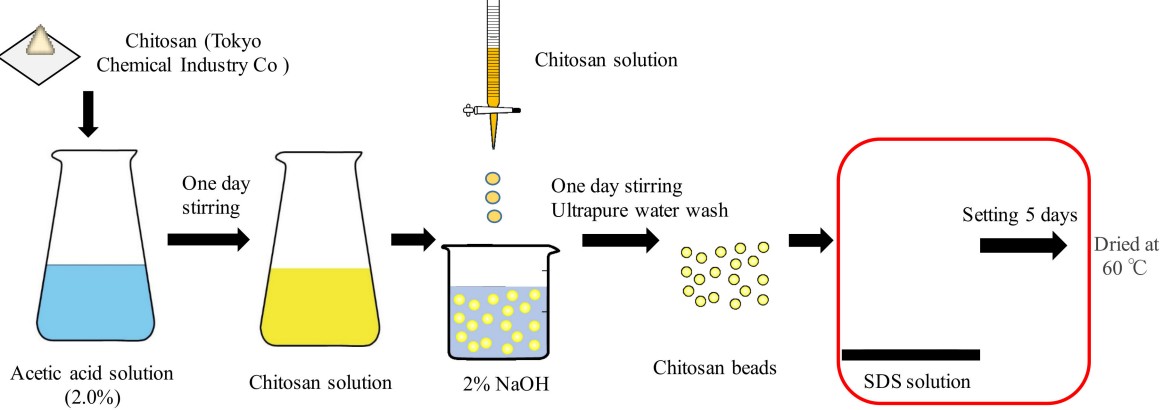

**Figure 1.** Procedure employed for the synthesis of sodium dodecyl sulfate (SDS)-chitosan beads.

### 2.2. Characterization of the Adsorbent

The diameter of chitosan beads was found to be about 0.5–2 mm (judging from 200 beads as representative chitosan beads). The data obtained after weighing 200 hydrogel beads showed that the dry weight per chitosan bead was $3.7 \times 10^{-4}$ g, which suggests that the adsorbent contained 98% moisture. Several characterization methods have been used to determine the physicochemical properties of pristine and modified chitosan. FT-IR spectra of the samples were recorded in the range of 4000–500 cm$^{-1}$ with a JASCO Japan FTIR-4200 spectrophotometer using the KBr pellet method. The surface morphology and element distribution of the chitosan beads before and after the adsorption of Cr(VI) were obtained using SEM-EDS (JEOL Japan: JCM-6000 with JED-2300). The surface chemistry properties of SDS-modified chitosan beads were investigated by X-ray Photoelectron Spectroscopy (XPS, Thermo Scientific Center: K-Alpha).

## 2.3. Adsorption Experiments

The SDS-chitosan beads synthesized in Section 2.1. were employed as the adsorbent of Cr(VI) in this study. The beads were placed in a 200 mL conical flask containing 50 mL of aqueous solution with a known amount of Cr(VI), and the suspensions were placed in a constant temperature-shaker (TAITEC Plus Shaker EP$^{-1}$ with Thermo Minder SX-10R) set at a prescribed temperature. Adsorption experiments were performed in the pH range of 1–7, SDS initial concentration of 10–9000 mg· L$^{-1}$, contact time of 1 to 72 h, adsorbent dosage of 0.01–0.06 g·L$^{-3}$, temperature of 288–318 K, and initial Cr(VI) concentration of 0.1–3.0 mg· L$^{-1}$. The pH of the test solution was adjusted by adding 0.1 mol·L$^{-1}$ NaOH or HNO$_3$. Secondary to the adsorption experiment, the suspension was filtered with 0.45 μm filter paper (Mixed Cellulose Ester 47 mm, Advantec MFS, Inc. (Tokyo, Japan)), and the concentration of Cr was determined by an ICP-MS (Thermo Scientific Center: X-series II). Replicate experiments were basically performed three times. The operating conditions of ICP-MS are shown in Table 1.

$$q_e = \frac{(C_i - C_e)}{m} \cdot V \tag{1}$$

where $q_e$ represents the adsorption capacities at equilibrium (mg·g$^{-1}$), $C_i$ and $C_e$ are the initial and equilibrium concentrations of Cr in a batch system, respectively (mg·L$^{-1}$); $V$ is the volume of the solution (L); $m$ is the weight of adsorbent (g).

**Table 1.** Operating conditions of inductively coupled plasma mass spectrometry (ICP-MS).

| Parameter | Condition |
| --- | --- |
| **Plasma Conditions** | |
| RF frequency | 27.1 MHz |
| RF power | 1400 W |
| **Gas conditions** | |
| Plasma gas flow | 15 L · min$^{-1}$ |
| Carrier gas flow | 1.2 L · min$^{-1}$ |
| **Sample conditions** | |
| Sampling depth | 6.5 mm |
| Sample uptake rate | 0.5 mL · min$^{-1}$ |
| Measurement point | 3 points/peak |
| Integration time | 1.0 sec/point |
| Measured isotope | $^{52}$Cr |

## 2.4. Adsorption Isotherm

Adsorption isotherms are commonly used to reflect the performance of adsorbents in adsorption processes. To examine the relationship between the metal uptake ($q_e$) and the concentration of metal ions ($C_e$) at equilibrium, adsorption isotherm models are widely employed for fitting the data. To investigate the equilibrium data, initial concentrations of metals were varied while the adsorbent weight of each sample was kept constant. Langmuir and Freundlich isotherms models were applied to evaluate the adsorption data obtained in this study.

The Langmuir model assumes monolayer adsorption on a surface and is given by

$$\frac{C_e}{q_e} = \frac{C_e}{q_{max}} + \frac{1}{K_L q_{max}} \tag{2}$$

where $C_e$ and $q_e$ are the concentration of Cr at equilibrium (mg·L$^{-1}$) and the amount of adsorption of Cr(VI) at equilibrium (mg·g$^{-1}$), respectively; $q_{max}$ is the maximum adsorption capacity on the surface of the chitosan bead (mg·g$^{-1}$); $K_L$ is the equilibrium adsorption constant (L·mg$^{-1}$). A plot of $C_e/q_e$ versus $C_e$ gives a straight line with a slope of $1/q_{max}$ and intercept of $1/(K_L q_{max})$.

On the other hand, the linearized Freundlich model isotherm is represented by the following equation:

$$\lg q_e = \lg K_F + (1/n)\,\lg C_e \qquad (3)$$

where $K_F$ is the adsorption capacity and $1/n$ indicates the adsorption intensity of the system. The plots of $q_e$ versus $C_e$ on a log scale can be plotted to determine the values of *1/n* and $K_F$ depicting the constants of the Freundlich model.

## 2.5. Kinetic Studies

Kinetic models have been proposed to determine the mechanism of the adsorption process, which provided useful data to improve the efficiency of the adsorption and feasibility of process scale-up. In the present investigation, the mechanism of the adsorption process was studied by fitting first-order and second-order reactions to the experimental data.

The pseudo-first-order model is given by the following equation:

$$\ln(q_e - q_t) = \ln(q_e) - k_1 t \qquad (4)$$

where $q_e$ and $q_t$ are the adsorption capacities of Cr using SDS-chitosan beads at equilibrium and time $t$, respectively ($mol \cdot g^{-1}$), and $k_1$ is the rate constant of the pseudo-first-order adsorption ($h^{-1}$).

The linear form of the pseudo-second-order rate equation is given as follows:

$$\frac{t}{q_t} = \frac{1}{kq_e^2} + \frac{t}{q_e} \qquad (5)$$

where $k$ is the rate constant of the pseudo-second-order adsorption ($g \cdot mol^{-1} \cdot h^{-1}$).

## 2.6. Adsorption Thermodynamics

Thermodynamic considerations of an adsorption process are necessary to investigate whether the process is spontaneous. Gibb's free energy change ($\Delta G^0$) is an indication of spontaneity of the chemical reaction. Gibb's free energy of the process can be determined by both standard enthalpy ($\Delta H^0$) and standard entropy ($\Delta S^0$) [39–42]. The free energy of an adsorption process is related to the equilibrium constant by the Van't Hoff equation:

$$\Delta G^0 = -RT \ln K_d \qquad (6)$$

where $R$ is the universal gas constant ($8.314 J mol^{-1}\,K^{-1}$) and $T$ is the temperature ($K$). The value of ln $K_d$ can be obtained by plotting ln $(q_e/C_e)$ vs. $q_e$ for the adsorption of metallic ions on SDS-chitosan and extrapolating $q_e$ to zero [43,44]. The thermodynamic parameters of the adsorption for the equations were also calculated by using the Langmuir constant ($K_L$) or Freundlich constants ($K_F$) instead of $K_d$.

$$\ln K_d = \frac{\Delta S^0}{R} - \frac{\Delta H^0}{RT} \qquad (7)$$

The slope and intercept of the Van't Hoff plot of $\ln K_d$ vs. *1/T* were used to determine the values of $\Delta H^0$ and $\Delta S^0$ based on equation (6). The plot of $\Delta G^0$ vs. $T$ can also give $\Delta H^0$ and $\Delta S^0$ by the following equation [43–45]:

$$\Delta G^0 = \Delta H^0 - T\Delta S^0 \qquad (8)$$

## 2.7. Effect of Competitive Anions ($Cl^-$, $NO_2^-$, $NO_3^-$, and $PO_4^{3-}$) on the Sorption of Cr(VI)

The effect of competitive anions on the sorption of Cr(VI) was studied in the following experiment. In this experiment, the initial concentration of Cr(VI) was taken as 1 mg$\cdot L^{-1}$. Each SDS-modified chitosan (1.0 g) bead was placed into a 200 mL conical flask contacting 50 mL of Cr(VI) solution in the presence of chloride ($Cl^-$), nitrite acid ($NO_2^-$), nitric acid ($NO_3^-$) and phosphate acid ($PO_4^{3-}$) ions at

different concentrations of 50, 100, and 200 mg· $L^{-1}$. Other experimental conditions (pH 4, contact time 24 h, and sorbent dosage 1.0 g·$L^{-1}$) and methods were essentially the same as those mentioned above in Section 2.3.

### 2.8. Desorption Experiments

From an industrial and technological point of view, it is desirable to recover and reuse the adsorbed material. A preliminary desorption experiment was conducted using SDS-chitosan beads after the adsorption of Cr(VI). The exhausted adsorbent was washed and dried overnight in each desorption experiment. After that, it was shaken for 24 h at 25 °C in a 100 mL flask which contained 50 mL of 0.1 M NaOH solution or ultrapure water. After the system attained equilibrium, the suspension was filtered, and the Cr(VI) content in the filtrate was determined finally.

## 3. Results and Discussion

### 3.1. Adsorption Experiment

#### 3.1.1. Effect of SDS Loading on the Adsorption of Cr(VI)

The maximum removal of Cr(VI) by modified chitosan beads can be obtained by controlling the SDS concentration. In order to research the optimum initial concentrations of SDS loading on chitosan beads for Cr(VI) removal, the chitosan beads were modified with SDS solutions with varying initial SDS concentrations (10–6000 mg/L). The experiment was conducted under the following conditions: adsorbent dose of 0.05g, initial concentration of Cr(VI) of 1 mg/L, pH of 4, and contact time of three days. The effect of surfactant loading on chitosan beads towards Cr(VI) adsorption is shown in Figure 2. The low SDS concentrations (0–1000 mg/L) range in Figure 2a was extended and shown in Figure 2b. From this figure, it was found that the adsorption capacity increased with an increase in the SDS concentration from 10 to 40 mg/L, and that it attained the maximum capacity at the initial SDS concentration of 40 mg/L. With a higher concentration of SDS, a higher adsorption capacity could not be obtained compared to the case of a low concentration range (10 to 100 mg/L). It is considered that the positively charged amino group was occupied by SDS, and thereby the adsorption reached saturation. Then, 40 mg/L was considered as the optimum SDS loading for further experiments.

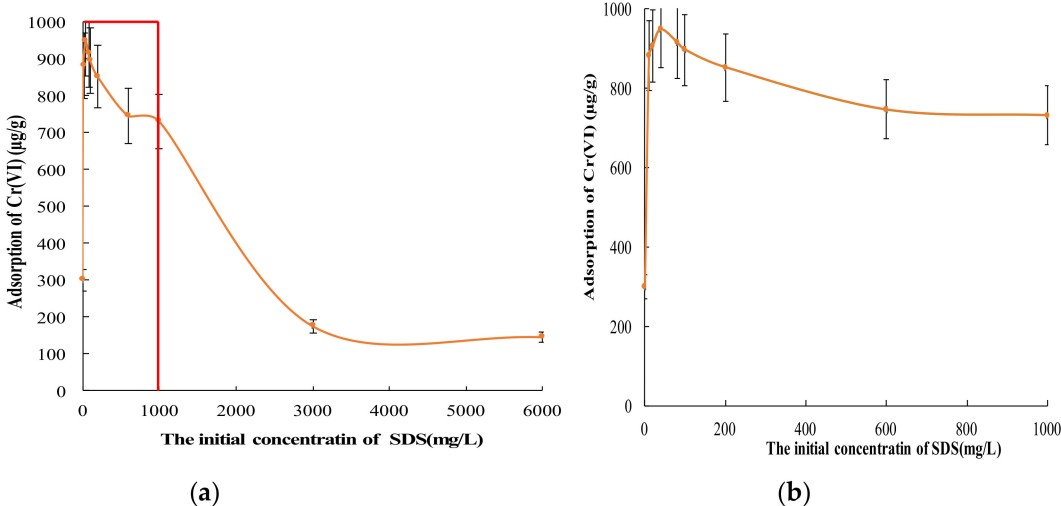

(a)  (b)

**Figure 2.** Adsorption of Cr(VI) for different initial SDS concentrations (10–6000 mg/L). (**a**) Effect of initial SDS concentrations (10–6000 mg/L) on the adsorption of chromium (Cr) (VI). (**b**) Effect of initial SDS concentrations (10–1000 mg/L) on the adsorption of Cr(VI).

### 3.1.2. Effect of pH on the Adsorption of Cr(VI)

　　The solution pH is one of the most important parameters in the adsorption process. Figure 3 shows the effect of the pH on Cr(VI) adsorption by SDS-modified chitosan beads. In this experiment, the shaking time was 24 h, temperature was 25 °C, dose of adsorbent was 0.02 g/L, and initial Cr(VI) concentration was 1 mg/L. The acidity of the solution had a significant effect on the adsorption of SDS-modified chitosan beads towards Cr(VI), where the amino groups of chitosan were protonated and positively charged. Moreover, the sulfate group is a typical strong acid group. Therefore, even under the acidic condition, which would not easily hydrolyze with water, surfactants could behave fully in their anionic form. It is well-known that chitosan molecules are protonated at amino groups which carry cationic adsorption sites and that they are dissolved in the acidic region [36]. Apart from cationic amino groups, chitosan chain adsorption (leading to interparticle bridging) can also occur through hydrogen bonds [46]. Similar to cationic polyacrylamide, chitosan is, thus, able to aggregate anionic soluble compounds (through electrostatic affinities) and simultaneously flocculate particulate matter through interparticle bridging mechanisms. However, the protonation of amino groups (i.e., the effective charge density of the chitosan polymer chain) can be influenced by the pH [47]. This makes it impossible to conduct adsorption experiments at pH levels of 1–3. The effect of pH on the adsorption of Cr(VI) by SDS-modified chitosan beads was estimated by adjusting the pH in the range of 4 to 10 (Figure 3). The maximum uptake of Cr(VI) ions took place at pH 4–5, which may be attributable to the changes in the surface charge of the adsorbent. With an increase of the pH at above pH 5, the uptake decreased.

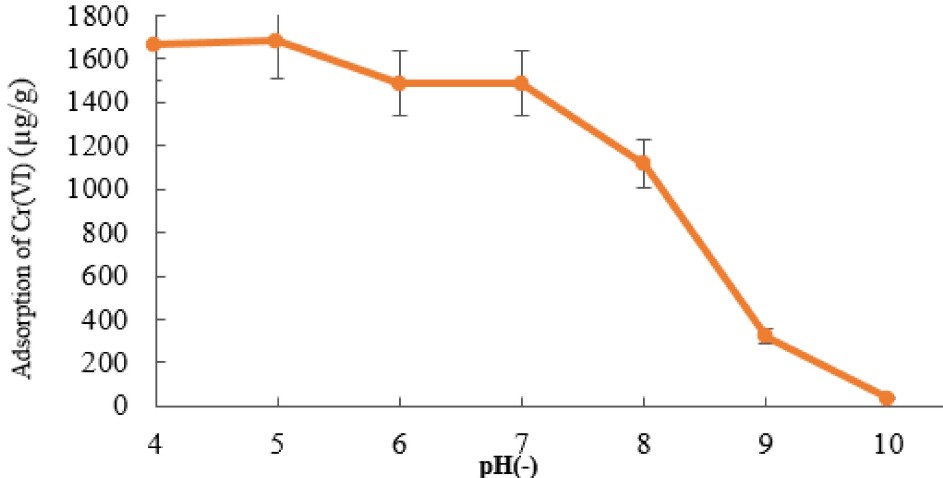

**Figure 3.** Effect of pH on the adsorption of Cr(VI) onto the SDS-modified chitosan beads.

　　Cr(VI) may be present in the form of $HCrO_4^-$ and $CrO_4^2$. As shown in Figure 4, $HCrO_4^-$ is dominant in the pH range of 2–4. However, $CrO_4^{2-}$ becomes increasingly dominant as the solution becomes more basic, and the form of $CrO_4^{2-}$ becomes stable above the pH of 7 [32]. The following equations indicate the shift process of Cr(VI) species:

$$HCrO_4^- \leftrightarrow CrO_4^{2-} + H^+ \quad pKa = 5.9 \tag{9}$$

$$H_2CrO_4 \leftrightarrow HCrO_4^- + H^+ \quad pKa = 4.1 \tag{10}$$

$$Cr_2O_7^{2-} + H_2O \leftrightarrow 2HCrO_4^- \quad pKa = 2.2 \tag{11}$$

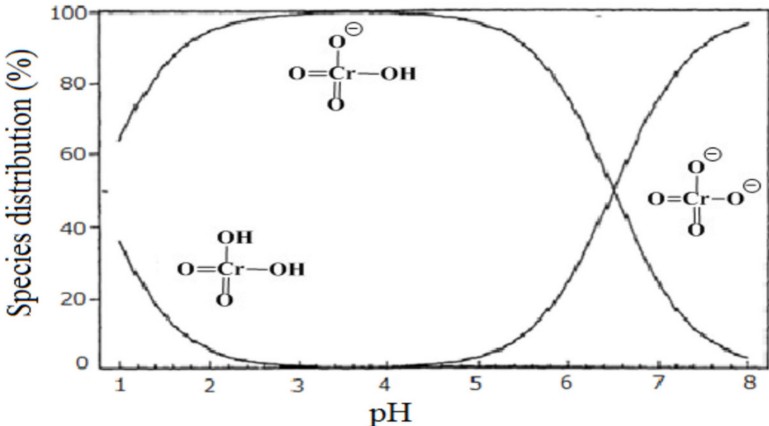

**Figure 4.** Species distribution curves of Cr(VI) in environmental water [32].

From these results, the functional groups combining with chromate ions may decrease due to the increase in their ion valence. It is supposed that amino acids with a positive charge combine with hydroxyl ions after binding with protons. Therefore, the adsorption capacity can largely depend on the kind of functional group and the existing form of Cr (VI). The surface of the SDS-chitosan beads became positively charged, owing to strong protonation in these pH ranges. This led to a stronger attraction between the positively charged surface and the negatively charged $Cr_2O_7{}^{2-}$ or $HCrO_4{}^-$.

### 3.1.3. Effect of Contact Time on the Adsorption of Cr(VI)

The effect of the contact time on the adsorption capacity of Cr(VI) by SDS-modified chitosan beads was explored. In this experiment, the concentration of Cr(VI) was set as 1 mg/L with the dose of 0.05 g at a temperature of 25 °C. The pH of the solution was kept at 4, in order to achieve the maximum removal of Cr(VI). Figure 5 demonstrates the adsorption of Cr(VI) with varying contact times from 1 to 96 h.

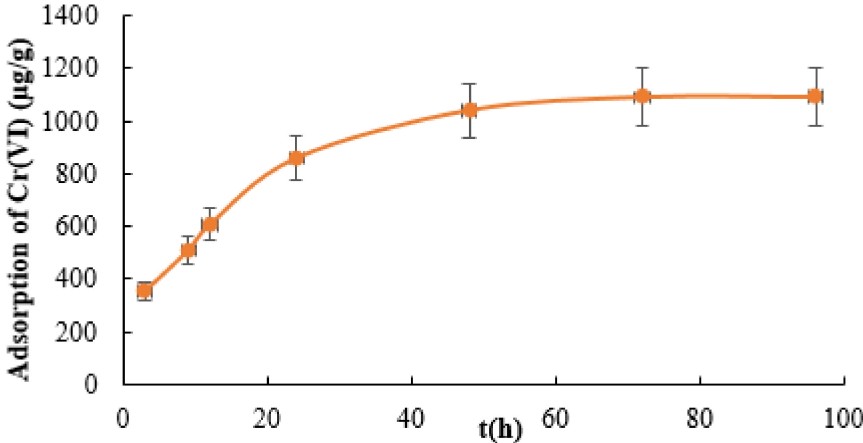

**Figure 5.** Effect of the contact time on the adsorption of Cr(VI) on the SDS-modified chitosan beads.

The adsorption capacity of SDS-chitosan beads for Cr(VI) increased sharply within the first 24 h, which may be attributable to the availability of the sites on the surface of the adsorbent. It is suggested that a concentration gradient is present in both the adsorbent and adsorbate in the solution [48]. Then, it reached adsorption equilibrium at 72 h, and afterwards, there was no appreciable increase (Figure 5). Hence, the optimized contact time was taken to be 72 h for further studies. The surface modification of the chitosan beads by SDS facilitates the adsorption through an ion-exchange mechanism [38]. Usually, the complexation mechanisms involve slower kinetics than the ion-exchange and hydrogen bonding reaction mechanism [49–51].

### 3.1.4. Effect of the Adsorbent Dosage on the Adsorption of Cr(VI)

To determine the effect of the adsorbent dosage on the removal of Cr(VI), the experiments were carried out by varying the dosage (from 0.4 to 1.0 mg/L) and keeping all other parameters constant (temperature: 25 °C; pH: 4; contact time: 24 h; initial concentration: 1.0 mg/L). The results are shown in Figure 6. The removal of more than 80% Cr(VI) was observed for the 0.8 mg/L dosage, but no notable increase was observed at a dosage of more than 0.8 mg/L. Therefore, 0.8 mg/L was regarded as the optimum dosage for the removal of Cr(VI) in this study. A higher dose provides a larger number of binding sites, which eventually causes the enhanced removal of Cr(VI).

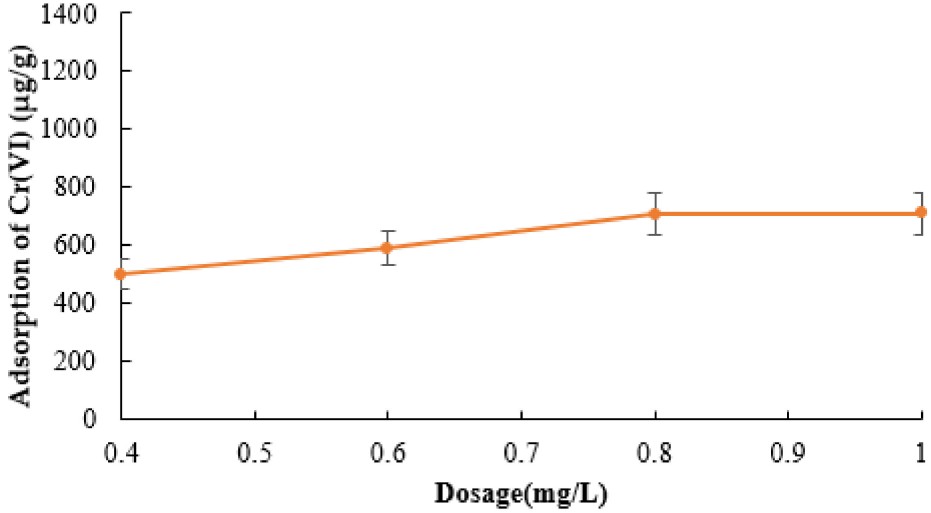

**Figure 6.** Effect of the adsorbent dosage on the adsorption of Cr(VI) on the SDS-modified chitosan beads.

### 3.1.5. Effect of Coexisting Ions on the Adsorption of Cr(VI)

The effect of competitive anions on the adsorption of Cr(VI) is shown in Figure 7. In this experiment, the initial concentration of Cr(VI) was set as 1 mg· $L^{-1}$. These counter ions were tested collectively, and all various ions were included at 50, 100, or 200 mg/L in solution. From this figure, the removal of Cr(VI) was remarkably decreased under the presence of common ions at above 50 mg· $L^{-1}$ (i.e., 50 times the Cr(VI) concentration or more), although no large decrease was observed when the concentration of each common ion was below 10 mg· $L^{-1}$ in our former preliminary experiments. Both Cr (VI) and other competitive anions may be attracted to the amino group by the electrostatic force. Therefore, Cr(VI) was shown to be inhibited by adsorption when the concentrations of coexisting ions were large.

### 3.1.6. Estimation of Partition Coefficient (PC)

Many adsorption studies have shown that the adsorption performance is usually evaluated and expressed by the maximum (or equilibrium) adsorption capacity. However, the maximum adsorption capacity is sensitively influenced by the initial concentration of the target pollutant (or expressing in more detail, what is left after the sorption reaction) [52,53]. When the sorbent is exposed to a higher concentration of objective targets, it is likely to exhibit a higher adsorption capacity. On the other hand, when the sorbent is exposed to lower levels of target species, it will show lower capacities. Therefore, in addition to the maximum adsorption capacity, it is effective to estimate using the concept of the partition coefficient (PC—adsorption capacity/final concentration) [53,54]. Comprehensive experimental data of the adsorption process under different effect factors are shown in Table 2. The data in Table 2 correspond to the data from Section 3.1.1. to Section 3.1.4. (i.e., Figures 2, 3, 5 and 6). From Table 2, it could be found that the adsorption affinity was fairly good under the following conditions when using either the concept of the adsorption capacity or that of the PC: pH of 4–5, contact time of 72 h, initial concentration of SDS 40 mg· $L^{-1}$, and adsorbent dose of 0.8–1.0 mg· $L^{-1}$.

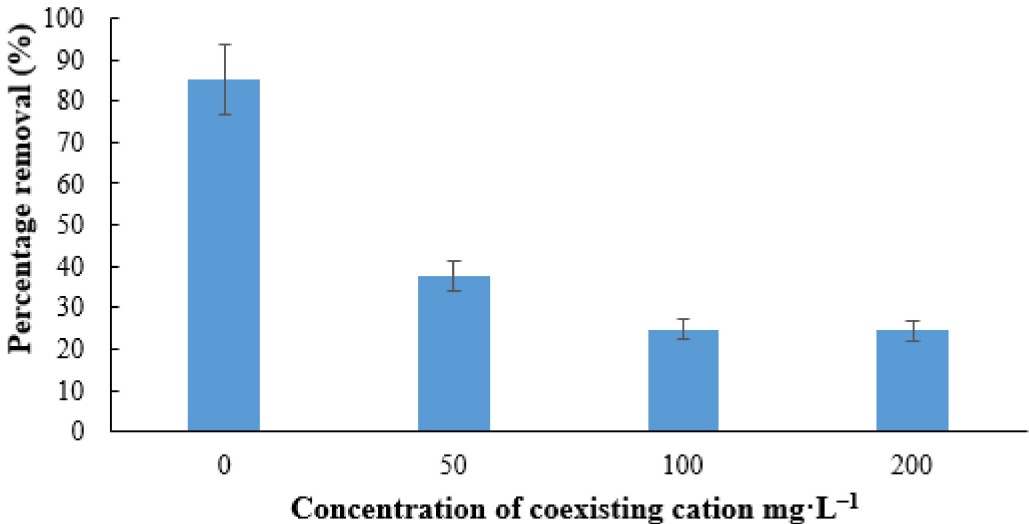

**Figure 7.** Effect of competitive anions on the adsorption of Cr(VI) on the SDS-modified chitosan beads.

**Table 2.** Detailed experimental data of the adsorption process under different effect factors.

| Target Ions | Adsorbent | Effect Factor | Final Concentration ($\mu g \cdot L^{-1}$) | Adsorption Capacity ($\mu g \cdot g^{-1}$) | Adsorption Capacity/ Final Concentration(PC) ($\mu g \cdot g^{-1} \cdot \mu M^{-1}$) |
|---|---|---|---|---|---|
| | | | 4 | 966.69 | 1665.39 | 0.034 |
| | | | 5 | 966.37 | 1681.62 | 0.035 |
| | | pH | 6 | 970.3 | 1484.99 | 0.031 |
| | | | 7 | 970.24 | 1488.23 | 0.031 |
| | | | 8 | 977.8 | 1114.81 | 0.022 |
| | | | 9 | 993.55 | 322.60 | 0.0065 |
| | | | 10 | 999.29 | 35.87 | 0.00071 |
| | | | 3 | 982.28 | 354.31 | 0.018 |
| | | Contact time (h) | 9 | 974.53 | 509.35 | 0.026 |
| Cr(VI) | | | 12 | 969.69 | 606.23 | 0.031 |
| | | | 24 | 957.04 | 859.12 | 0.045 |
| | SDS | | 48 | 948 | 1039.91 | 0.055 |
| | | | 72 | 945.51 | 1089.86 | 0.058 |
| | | | 96 | 945.50 | 1090.21 | 0.058 |
| | | | 0 | 985 | 300 | 0.015 |
| | | Initial | 10 | 955.93 | 881.49 | 0.046 |
| | | | 20 | 954.67 | 906.69 | 0.047 |
| | | | 40 | 952.63 | 947.37 | 0.050 |
| | | Concentration of SDS ($mg \cdot L^{-1}$) | 80 | 954.25 | 915.02 | 0.048 |
| | | | 100 | 955.20 | 896.11 | 0.047 |
| | | | 200 | 957.41 | 851.81 | 0.044 |
| | | | 600 | 962.73 | 745.33 | 0.039 |
| | | | 1000 | 963.44 | 731.16 | 0.038 |
| | | | 0.4 | 800 | 500 | 0.4 |
| | | Dose ($mg \cdot L^{-1}$) | 0.6 | 646.6 | 589 | 0.86 |
| | | | 0.8 | 434.4 | 707 | 1.93 |
| | | | 1.0 | 292 | 708 | 2.42 |

### 3.2. Characterization of SDS Modified Chitosan

The SEM-EDS images of chitosan beads and SDS-chitosan beads (40 mg/L SDS loading) are shown in Figure 8. From SEM images, it could be determined that the diameter of these beads was roughly 800 μm. These figures show that the surface of the membrane appears concave and convex. This may be attributable to the loss of water (about 98%) contained in the sorbent upon the drying of chitosan. It is considered that adsorption proceeded with physical and chemical adsorption. From the mapping images, it is shown that Cr ions were sufficiently adsorbed on the adsorbent.

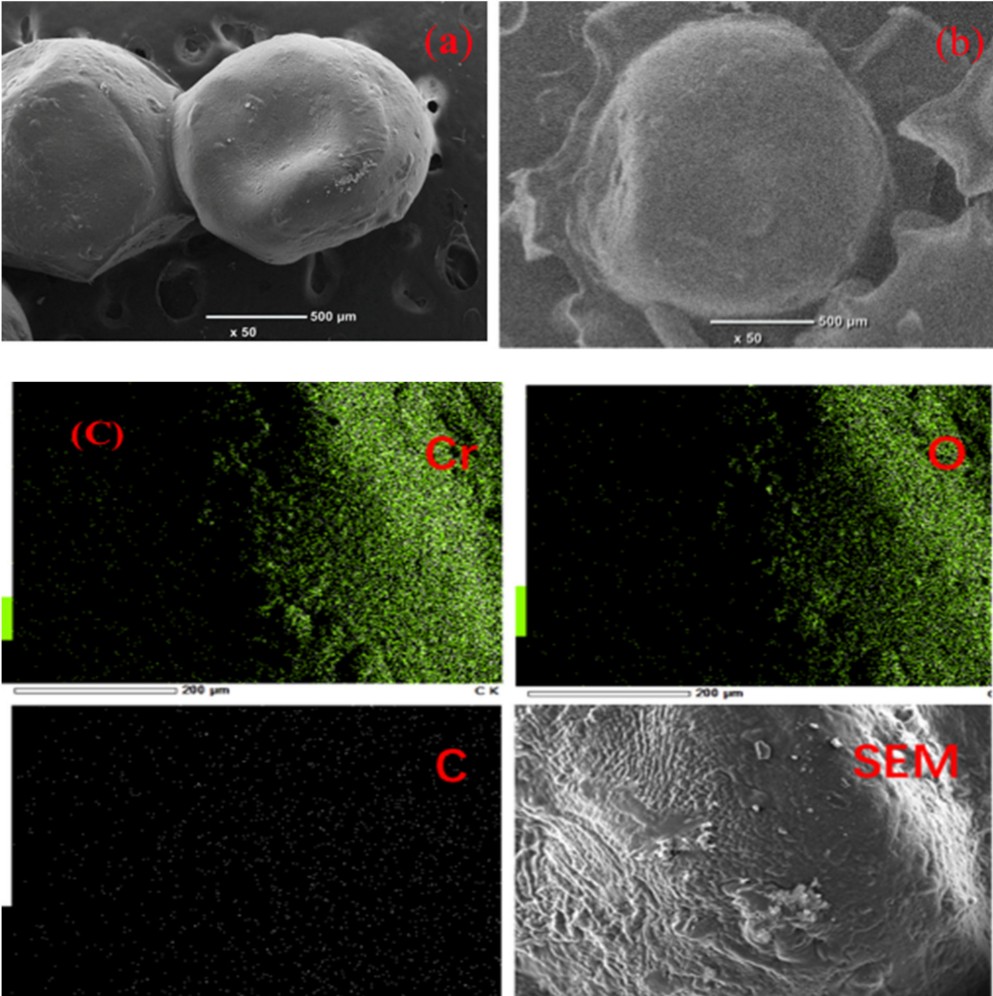

**Figure 8.** Scanning electron microscopy (SEM) images of (**a**) chitosan beads and (**b**) SDS-modified chitosan beads, and (**c**) mapping image of SDS-modified chitosan beads after the adsorption of Cr(VI).

The FT-IR spectra of the chitosan and SDS-chitosan beads are exhibited in Figure 9. The broad and intense peak at 3400 to 3500 cm$^{-1}$ corresponds to O-H and -NH$_2$ stretching vibrations of hydroxyl groups of chitosan and SDS-chitosan [55–57]. The peak at 2871 cm$^{-1}$ is related to the aliphatic methylene group [58,59]. The wide peaks at 1560 to 1640 cm$^{-1}$ and 1110 cm$^{-1}$ show the imine group and ether group [57]. The SDS-modified chitosan bands at around 1248 cm$^{-1}$ are characteristic of the asymmetrical vibrations of the C-O-S group [59,60]. The adsorbent could be identified as a composite of SDS and chitosan.

XPS analysis was employed to survey the chemical compositions and binding condition of the surface on the samples. Chitosan beads with different amounts of SDS loading were analyzed (Figure 10a—Chitosan; Figure 10b—SDS100-chitosan; Figure 10c—SDS600-chitosan; Figure 10d—SDS6000-chitosan). As can be seen in Figure 10, the C1s spectra of these samples displayed peaks at 284.5 eV and 286.5 and 288.5 eV, corresponding to C-C bonds C-O and C=O, respectively. It also shows that the S 2p spectra of SDS600-chitosan and SDS6000-chitosan displayed peaks at 169 eV. From Table 3, it could be found that the predominant elements were carbon and oxygen, and that the SDS100-chitosan and SDS600-chitosan atomic values of the N element are 1.75% and 1.69%, respectively. The SDS600-chitosan and SDS6000-chitosan atomic values of the S element are 2.78% and 6.85% respectively, which suggests that the higher the concentration of loading SDS is, the higher the concentration of the S element is.

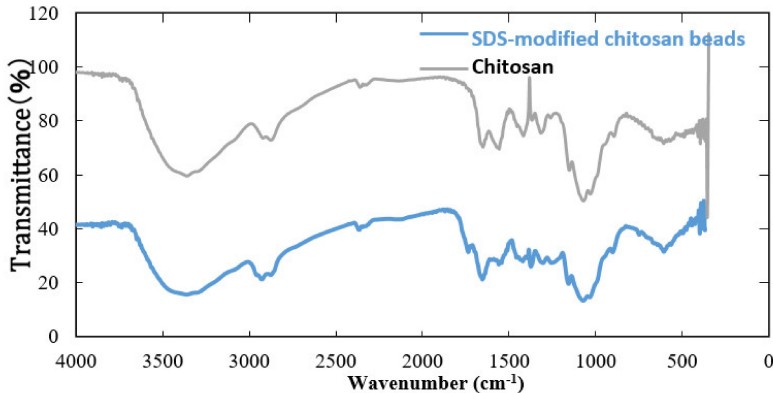

**Figure 9.** Fourier transform-infrared spectroscopy (FT-IR) spectra of chitosan and SDS-modified chitosan beads.

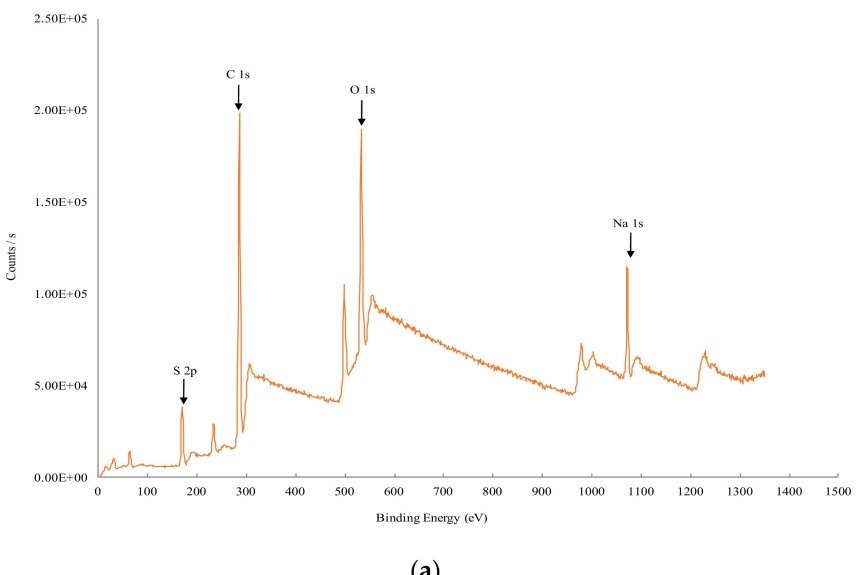

(**a**)

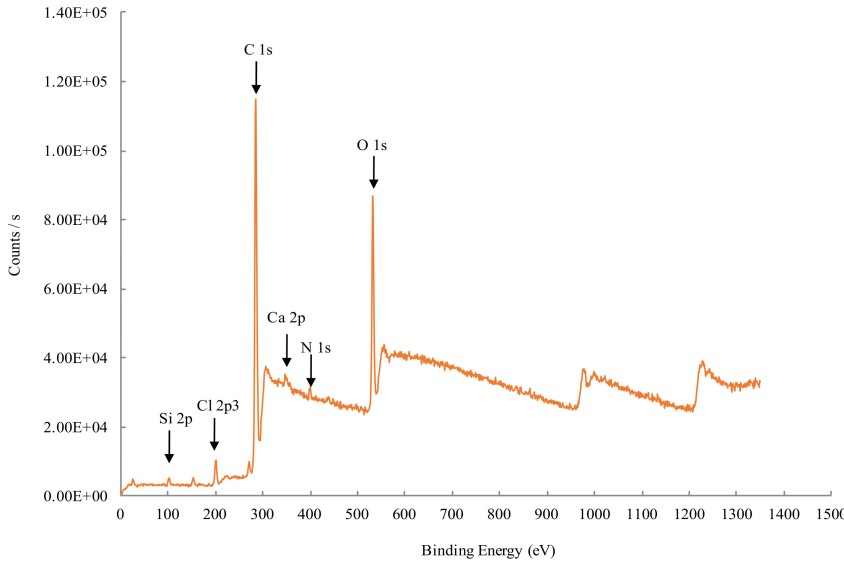

(**b**)

**Figure 10.** *Cont*.

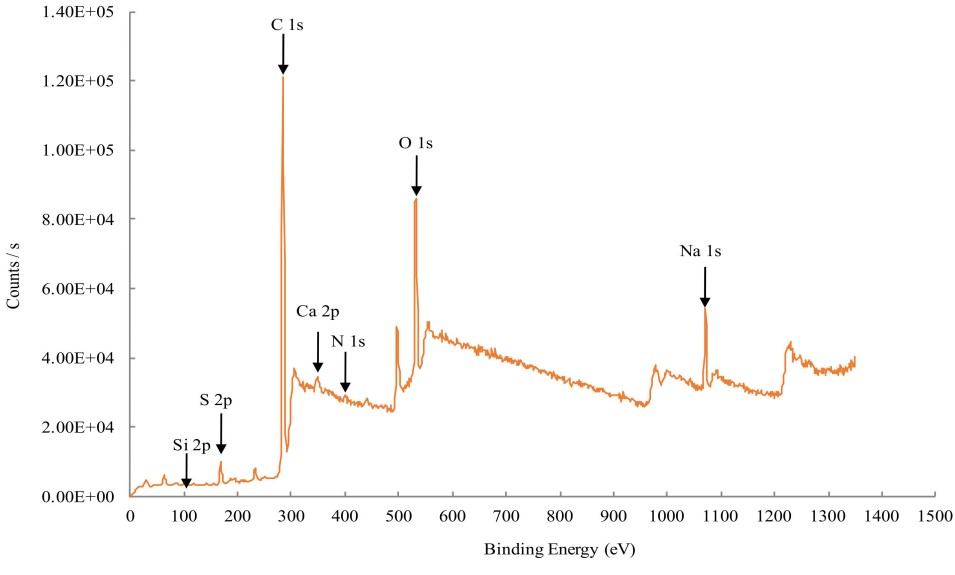

(**c**)

(**d**)

**Figure 10.** X-ray photoelectron spectroscopy (XPS) spectra of (**a**) chitosan, (**b**) SDS100-chitosan, (**c**) SDS600-chitosan, and (**d**) SDS6000-chitosan beads.

**Table 3.** Atomic ratio of each chitosan bead obtained by XPS analysis.

| Name Atomic% | Chitosan | SDS100 -Chitosan | SDS600 -Chitosan | SDS6000 -Chitosan |
|---|---|---|---|---|
| Si 2p | 4.41 | 2.06 | 0.81 | |
| S 2p | | | 2.78 | 6.85 |
| C 1s | 77.1 | 76.52 | 75.91 | 67.66 |
| Cl 2p3 | 3.94 | 3.55 | | |
| Ca 2p | 0.27 | 0.33 | 1.1 | |
| N 1s | | 1.75 | 1.69 | |
| O 1s | 14.27 | 15.78 | 14.34 | 20.19 |

### 3.3. Adsorption Isotherms

Adsorption isotherms of Cr(VI) on SDS-chitosan beads were studied with varying initial concentrations from 0.010 to 3.0 mg/L, under optimized conditions in terms of the pH (pH 4), contact time (72 h), and dosage of the adsorbent (0.8 mg/L) at 298 K in this work. Adsorption isotherms are generally used to reflect the performance of adsorbents in adsorption processes. Herein, two common adsorption models, consisting of Langmuir [61,62] and Freundlich [61,63] equations, were employed to explain the adsorption of Cr by SDS-chitosan beads (Figure 11). The adsorption data acquired for Cr(VI) using SDS-chitosan beads were investigated by Langmuir and Freundlich equations, and the results are shown in Figures 12 and 13, respectively. All isotherm parameters calculated from the two models are listed in Table 4, along with the correlation coefficients ($R^2$). High correlation coefficients indicate that Cr(VI) sorption can be well-described by the Langmuir and Freundlich isotherms. In particular, the Langmuir model was preferable, and the maximum adsorption capacity was estimated to be 3.23 mg/g. The Langmuir isotherm model is known to elucidate the monolayer adsorption on homogeneous surfaces. The results implied that the adsorption of Cr(VI) was a monolayer coverage process [64]. This indicated a strong potential in the application of SDS-chitosan beads for Cr(VI) removal from an aqueous phase. On the other hand, it was found that the $R^2$ value obtained from the Freundlich model was not small, and favorable adsorption was suggested, judging from the value of $1/n$ [65]. The isotherm parameters revealed that the SDS-chitosan bead with special structures could efficiently enhance the adsorption capacities.

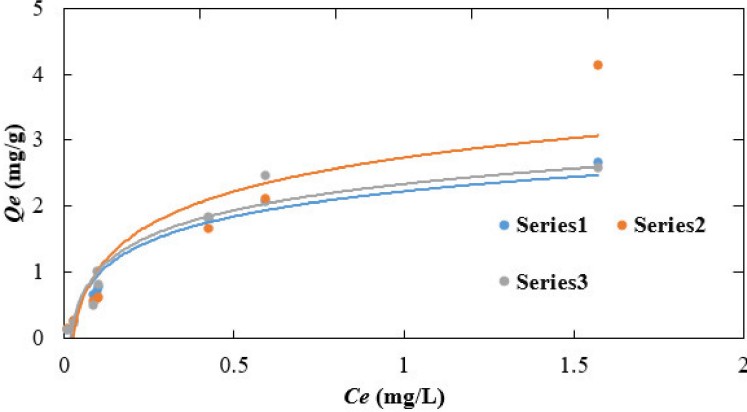

**Figure 11.** Adsorption isotherms of Cr(VI) using the SDS-chitosan beads.

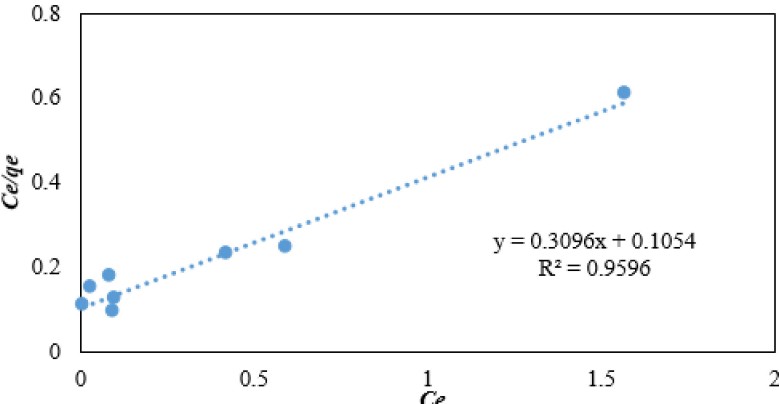

**Figure 12.** The Langmuir isotherm of Cr(VI) adsorption on the SDS-chitosan beads.

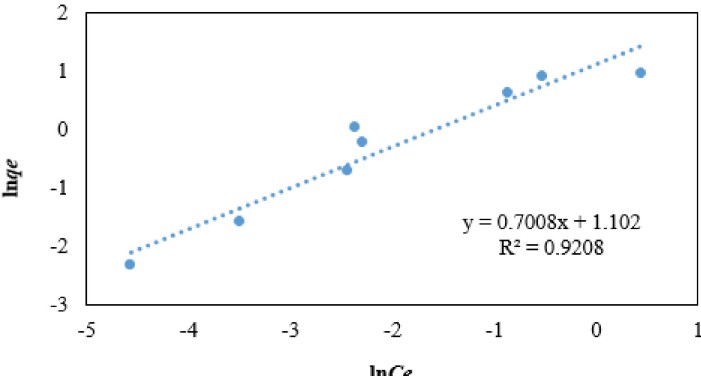

**Figure 13.** The Freundlich isotherm of Cr(VI) adsorption on the SDS-chitosan beads.

**Table 4.** Isotherm parameters for Cr(VI) adsorption onto the SDS beads.

| Metal | T (°C) | Langmuir Isotherm | | | Freundlich Isotherm | | |
|---|---|---|---|---|---|---|---|
| | | $Q_{max}(mg/g)$ | $R_L$ | $R^2$ | $K_F(mg/g)$ | $1/n$ | $R^2$ |
| Cr(VI) | 25 | 3.23 | $0.308 \times 10^{-4}$ | 0.960 | 3.01 | 0.700 | 0.921 |

*3.4. Kinetic Studies*

An adsorption kinetics study was conducted to explore the relationship between the adsorption amount $q_t$ and time $t$. According to Figure 14, the adsorption content of Cr(VI) by SDS-chitosan beads increased significantly within 96 h. The quick adsorption within the initial 24 h indicated that uptake of Cr(VI) was mainly caused by chemical sorption or surface complexation. This might be associated with the abundant exposure of sorption sites on the adsorbent surface. As the sorption sites were gradually occupied by Cr(VI), the adsorption rate of Cr(VI) became slower with a lapse of time and ultimately, approached equilibrium [66,67].

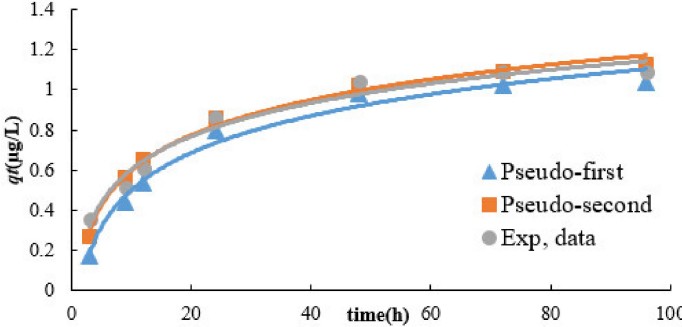

**Figure 14.** Pseudo-first-order and pseudo-second-order plot of Cr(VI) adsorption on the SDS-chitosan beads.

For the purpose of comprehending the adsorption kinetics in more detail, the pseudo-first-order, pse For the purpose of comprehending the adsorption kinetics in more detail, the pseudo-first-order, pseudo-second-order, and intraparticle diffusion kinetic models were applied to explain the kinetic sorption process [68,69]. The fitting curves of pseudo-first-order and pseudo-second-order models are presented in Figures 15 and 16, respectively. The calculated parameters, along with the correlation coefficient ($R^2$) of these models, are presented in Table 5. The adsorption kinetics based on the experimental values were both in good agreement with the pseudo-first-order and pseudo-second-order kinetic models judging from the high correlation coefficients. The fitness of the pseudo-first-kinetic model implied that the rate-controlling step might involve chemisorption or chemical bonding between Cr(VI) and the functional groups of adsorbents. On the other hand, the rapid phase in the initial step

of the adsorption process may imply physical adsorption or exchange at the adsorbent surface [61,70]. Although the adsorption data could be well-described by the pseudo-first-kinetic model, the diffusion of Cr(VI) into pores could play an important role for Cr(VI) adsorption on the adsorbent, since SDS-chitosan beads are porous structures. Therefore, an intraparticle diffusion model was also applied to elucidate the diffusion mechanism and to investigate whether the film or pore diffusion was the controlling step in the adsorption process. The plots of $q_t$ versus $t^{1/2}$ for the adsorption of Cr(VI) fitting as an intraparticle diffusion model are provided in Figure 17. From this figure, it is revealed that plural processes influence the adsorption process for the adsorption of Cr(VI) by SDS-chitosan beads. In the pseudo-first-kinetic model, the adsorption rate was very high. This may be attributed to the film diffusion of Cr(VI) through the hydrodynamic layer to the surface of SDS-chitosan beads and the diffusion of Cr(VI) through the boundary layer to the external surface of the adsorbent. As the sorption on the external surface reaches saturation, Cr(VI) can enter into the pores of the SDS-chitosan beads and be adsorbed on the internal surface of the mesopores. It is considered that the intraparticle diffusion starts to fall down and reaches an equilibrium stage with the lowering of the Cr(VI) concentration in solution [61,71].

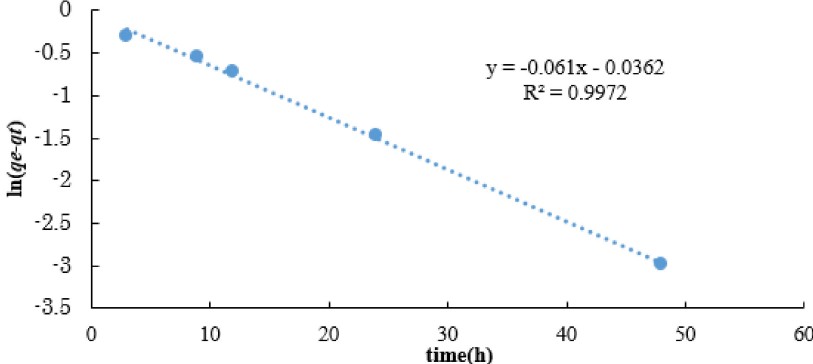

**Figure 15.** Pseudo-first-order linear kinetic model of Cr(VI) adsorption on the SDS-chitosan beads.

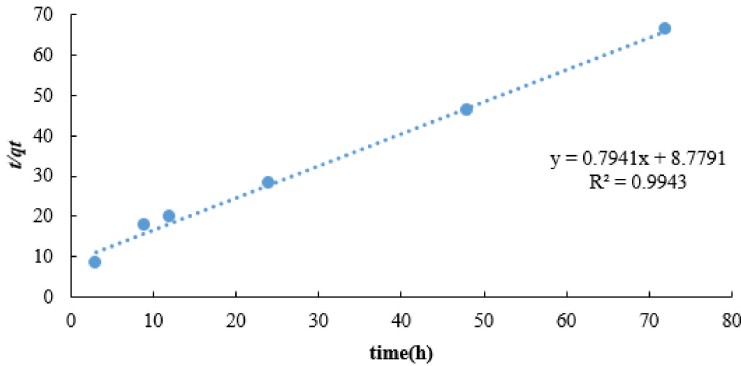

**Figure 16.** Pseudo-second-order linear kinetic model of Cr(VI) adsorption on the SDS-chitosan beads.

**Table 5.** Kinetic parameters of Cr(VI) adsorption onto the SDS-chitosan beads.

| Adsorbent | $q_e$ (P-mg/g) | Pseudo-First-Order Model | | | Pseudo-Second-Order Model | | |
|---|---|---|---|---|---|---|---|
| | | $q_e$ (P-mg/g) | $k_1$ ($min^{-1}$) | $R^2$ | $q_e$ (P-mg/g) | $k^2$ ($g/mg \cdot min^{-1}$) | $R^2$ |
| SDS40 | 1.09 | 1.04 | 0.0610 | 0.997 | 1.26 | 0.0718 | 0.994 |

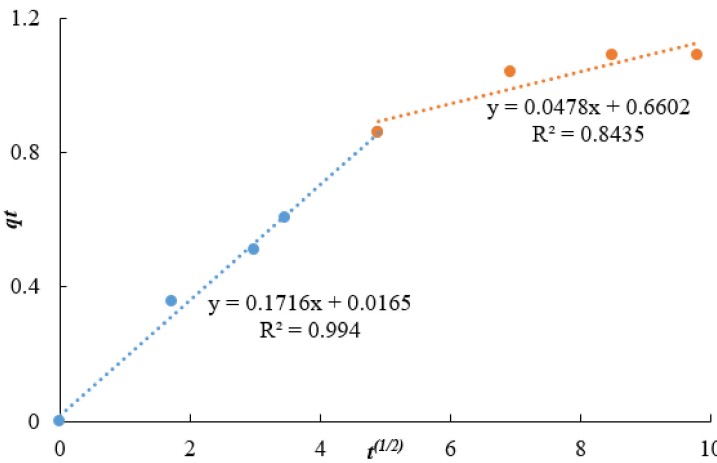

**Figure 17.** Intraparticle plot for the adsorption of Cr(VI) by the SDS-chitosan beads.

### 3.5. Thermodynamic Study

The adsorption experiments of Cr(VI) were carried out from 288 to 318 K for the thermodynamic investigation. Thermodynamic parameters such as Gibb's free energy change ($\Delta G$), enthalpy change ($\Delta H$), and entropy change ($\Delta S$) were calculated from the adsorption isotherms at different temperatures. All of the thermodynamic parameters for the adsorption of Cr(VI) on SDS-chitosan beads are tabulated in Table 6, and the adsorption capacity of Cr(VI) depending on the temperature is shown in Figure 18. The negative values of $\Delta G$ suggest the preferred occurrence of spontaneous adsorption over the tested temperatures (288 to 318 K). As shown in Figure 18, the adsorption capability of Cr(VI) on SDS-chitosan beads increased with an increase in temperature. In addition, the minus value of $\Delta G$ increased with the increase in temperature. This indicates that the adsorption of SDS-chitosan beads on Cr(VI) is more favorable at a higher temperature. Besides, the enthalpy $\Delta H$ of SDS-chitosan beads ($\Delta H > 0$) denoted the endothermic adsorption reaction, with the probable better adsorption results at a high temperature. It is known that the value of $\Delta G$ is between 0 and $-20$ kJ/mol for physisorption, and that it is between $-80$ and $-400$ kJ/mol for chemisorption [72]. In certain conditions, the physisorption and chemisorption can be classified by the magnitude of $\Delta H$ and $\Delta G$. Bonding strengths of $< 84$ kJ/mol are typically considered as those of physisorption interaction [73]. Then, it is suggested that the adsorption of Cr(VI) on SDS-chitosan beads can be mainly dominated by physisorption judging from the values of $\Delta H$ and $\Delta G$.

**Table 6.** Thermodynamic parameters for the adsorption of Cr(VI) on SDS-modified chitosan beads.

| T(K) | $\Delta H$(kJ/mol) | $\Delta S$(J/mol) | $\Delta G$(kJ/mol) |
|------|------|------|------|
| 288 | 80.70 | 288.18 | $-2.34$ |
| 298 | - | - | $-5.22$ |
| 308 | - | - | $-8.10$ |
| 318 | - | - | $-10.98$ |

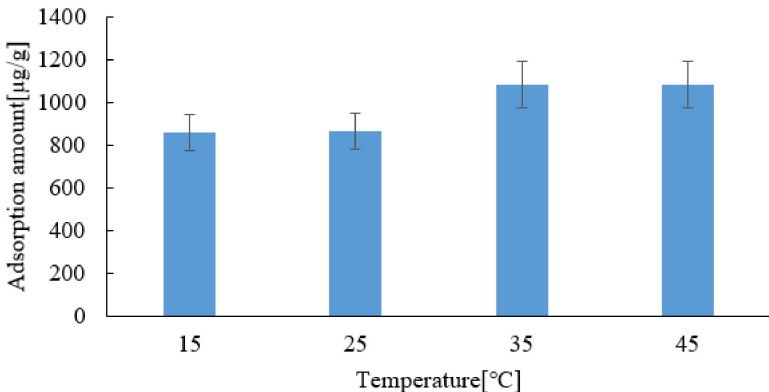

**Figure 18.** Effect of temperature on the adsorption of Cr(VI) on the SDS-modified chitosan beads.

### 3.6. Desorption Study

In terms of economically viable applications, the adsorbed material is supposed to be recovered and reused. Based on the procedure of 2.8, preliminary desorption experiments were conducted by using 0.1 mol/L NaOH or pure water, and the results are presented in Figure 19. From this figure, the desorption efficiency of Cr(VI) was found to be 50% by using 0.1 mol/L NaOH. On the other hand, the desorption was considerably lower with $H_2O$ than NaOH. It is suggested that NaOH can be a desorption agent for Cr(VI), although further investigation is needed for the effective recovery and recycling of Cr (VI).

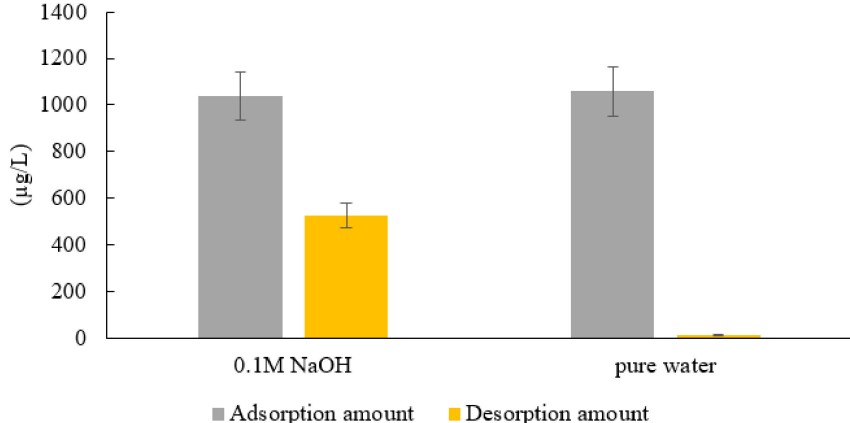

**Figure 19.** Desorption efficiency of Cr(VI) using NaOH or pure water.

### 3.7. Comparison with Other Adsorbents

The maximum adsorption capacity and partition coefficient (PC) of Cr(VI) on SDS-chitosan beads were compared with those of different adsorbents reported in the literature. The comparison is summarized in Table 7. It is noteworthy that the PC values of SDS-chitosan beads used in this work are remarkably high, although the adsorption capacity is not so high compared to other adsorbents. As denoted in 3.1.6., the maximum adsorption capacity can be sensitively influenced by the initial concentration of objective targets. We regarded that the concept of PC was also important when comparing the adsorption affinity of different adsorbents.

**Table 7.** The comparison of adsorption properties of several adsorbents.

| Adsorbents | Final Concentration (mg·L$^{-1}$) | Adsorption Capacity (mg·g$^{-1}$) | Partition Coefficient (mg·g$^{-1}$·mM$^{-1}$) | Reference |
|---|---|---|---|---|
| Magnetic Chitosan | 30.6 | 69.4 | 2.3 | [14] |
| Carboxymethyl Chitosan-Silicon Dioxide | 39.9 | 80.7 | 2.1 | [30] |
| Chitosan-g-poly/silica | $44.3 \times 10^{-3}$ | $55.7 \times 10^{-3}$ | 1.3 | [74] |
| Crosslinked chitosan bentonite composite | 210.87 | 89.1 | 0.42 | [75] |
| Graphene oxide/chitosan | 13.8 | 86.2 | 6.2 | [76] |
| Ethylenediamine-magnetic chitosan | 148.2 | 51.8 | 0.35 | [77] |
| SDS-chitosan | 0.34 | 3.23 | 9.5 | This study |

## 4. Conclusions

In this study, chitosan was chemically modified by sodium dodecyl sulfate (SDS) to enhance its adsorption capacity for the removal of Cr(VI). The SDS-chitosan beads were characterized by SEM-EDS, FT-IR, and XPS. The effect of several operating parameters, such as the loading amounts of SDS, solution pH, contact time, adsorbent dose, temperature, and initial Cr(VI) concentration, on the adsorption performance was examined in a batch system. The experimental data were found to be fit best using a Langmuir isotherm and pseudo-first-order kinetic models. The performance of the adsorption process was favorable at pH 4–5 and higher temperatures. The results showed that the maximum adsorption capacity and partition coefficient (PC) of Cr(VI) on SDS modified chitosan beads were 3.23 mg·g$^{-1}$ and 9.5 mg·g$^{-1}$·mM$^{-1}$, respectively. In conclusion, the SDS-chitosan beads synthesized in this work can be effectively utilized for removing Cr(VI) ions.

**Author Contributions:** Experiment and writing, X.D.; experiment and data evaluation, C.K.; measurements of instrument, H.Z.; measurements of instrument and data evaluation, N.M.; supervision and writing, N.K. All authors have read and agreed to the published version of the manuscript.

**Funding:** The present study was partially supported by The Uchida Energy Science Promotion Foundation. This research was also supported by a fund for the promotion of Niigata University KAAB Projects from the Ministry of Education, Culture, Sports, Science and Technology, Japan.

**Acknowledgments:** The authors are grateful to Haruo Morohashi of the Industrial Research Institute of Niigata Prefecture for the measurement of XPS and useful advice. The authors also thank Ohizumi, M. of the Office for Environment and Safety, and Teraguchi, M., Nomoto, T. and Tanaka, T. of the Facility of Engineering in Niigata University for permitting the use of ICP-MS, FT-IR, and SEM-EDS.

**Conflicts of Interest:** The authors declare no conflicts of interest.

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
