# Peer review of "Removal of Chromium(VI) by Chitosan Beads Modified with Sodium Dodecyl Sulfate (SDS)"

_applsci, doi:10.3390/app10144745_

Round 1

Reviewer 1 Report

Synopsis:

The authors investigated the use of synthesized chitosan beads modified with sodium dodecyl sulfate (SDS) to sorb Cr(VI) from solution.  The authors investigated the impacts of SDS loading, Cr(VI) initial concentration, contact time, pH, and the presence of competitive anions on chromate sorption.  The authors also investigate the reaction parameters and mechanisms using thermodynamic and kinetic modeling.  Particle investigation was done via SEM-EDS, XPS, and FTIR, while Cr(VI) concentrations were determined by ICP-MS. 

General Comments:

The paper is interesting, but the grammar could be improved (some specific examples below).

One concern I have about this work is that I see no mention of replicate experiments being done.  Nothing about it in the methodology, and nothing in the figures/ figure captions to indicate that the data points represent anything other than a single measurement.

Also, I’m a little unclear on the exact procedures used in these experiments.  The text in section 2.1states that the chitosan beads are equilibrated with SDS to create the modified beads used in the sorption experiments (the rest of the text implies this as well).  However, the text in section 2.3 implies that in the sorption experiments, the SDS chitosan beads are in a solution contain Cr(VI) and (more?) SDS.  Is that the case?  If so, why pre-treat with SDS?  If SDS is also in the experiments how does it contribute to Cr(VI) removal independent of the beads?  Please clarify this.

Was anything done to look at Cr(VI) reduction in these experiment?  I haven’t worked with chitosan or SDS, but I do know that at pH 4 dissolved organic matter can reduce Cr(VI) to Cr(III).

Specific Comments:

Line 20: States that the SDS chitosan beads have a high affinity for Cr(VI), but line 431 says that the adsorption capacity is low compared to other adsorbents.  Is this a rate vs. amount issue? 

Line 52:  What do you mean by “no second pollution”?  Are you saying there’s no secondary pollution from synthesizing SDS chitosan beads?  Or are you saying that after sorption of Cr(VI) by the beads there is no further waste stream?  How is the later possible, the Cr has to go somewhere.

Line 92:  What size are these beads?  You have SEM images that indicate ~800 nm in diameter, is this the typical size?

Line 189-190 & Figure 1:  The text indicates that the low concentration portion of figure 1a (0-1000 mg/L) was extended and shown in figure 1b.  So why does the data in figure 1 b look so different from the data in figure 1a?  Figure 1a indicates a Cr(VI) sorption maximum in the 800-900 mg/L SDS range, whereas figure 1b has a maximum at 40 mg/L.  The trend in the data is also different.  What’s going on with these figures?

Line 223-225:  The text that begins with “At lower pH range, …”  is unclear to me.  The sentence states that adsorption of metal ions is hindered by positive surface charges at low pH, and then states that adsorption is favored at high pH due to electrostatic attractions.  While this is true of metal cations, it is not true of the Cr(VI) oxy-anion.  Since the entire paragraph was discussing the sorption of Cr(VI), and figure 2 clearly shows the sorption of Cr(VI) decreasing as pH increases, I don’t understand what point the authors were trying to make with these two lines of text.

Line 246-256 & Figure 5:  This section deals with the impact of initial Cr(VI) concentration on the amount of Cr sorption by the SDS chitosan beads.  It states that 0.8 mg/L is the ideal dosage with more than 80% being sorbed, and no increase in the amount sorbed at higher initial concentrations.  However, figure 5 doesn’t seem to support this.  The y-axis is labeled “Concentration after adsorption” which would be how much Cr(VI) is still in solution, which is nearly all of it, indicating a very small percentage being sorbed.  Is the y-axis of figure 5 supposed to be the amount of Cr(VI) sorbed?  If so, why is it in units of concentration?  Regardless, the figure has issues at low concentrations.  At a dosage of 0.6 mg/L there is 0% adsorption (or 100% if the y-axis is mislabeled), while at a dosage of 0.4 mg/L there is more Cr(VI) left in solution (or adsorbed if the axis is mislabeled) than was added to the experiment in the first place.  Please clarify what is going on in this section.

Section 3.1.5 & Figure 6:  In the methodology it says that sorption experiments were conducted with the chloride, nitrite, nitrate, and phosphate counter ions at a concentration of 50, 100, or 200 mg/L.  What I’m not clear on is whether these counter ions were tested independently, or collectively.  If individually, then why is this not reflected in the discussion and figure 6.  If they were tested collectively, what was the concentration of the various counter ions?  Chloride is a very different counter ion in terms of interacting with other components of the aqueous system than phosphate.  Why report the results as a collective?  Is this collection of counter ions supposed to be representative of a particular aqueous environment? 

Line 261:  remove the word “not”

Line 282 Table 2:  Where is this data coming from?  The Final Concentration numbers do not seem to be consistent with the Figure 2 sorption vs pH numbers.  Is this a separate batch of experiments?  If so, what are the conditions?

Line 422 & Figure 18:  Figure 18 shows that very little Cr(VI) was desorbed with water, so the text in lines 421 & 422 needs to be changed to reflect that desorption is considerably lower with H2O rather than NaOH.

Line 423: mentions the need to investigate recovery/recycling of Cr(VI), possibly contradicts line 52 statement about “no second pollution”.

Reviewer 2 Report

General comment: The authors produced beads used as an adsorbent to remove Cr(VI). Among several materials and other separation methods (e.g. precipitation), those biosourced functionalized beads are considered as sustainable technic to removed Cr(VI), and potentially other heavy metals. I recommend this manuscript to be accepted in Applied Sciences, but major revision and clarifications concerning the removal mechanisms are required before publication.

Specific comments:

Lines 53-54: Please check and/or rephrase ‘’ which is the most abundant biopolymer in nature originated from cellulose’’. Cellulose could be considered as a biopolymer itself and, de facto, the most abundant.

Line 87: What are the chitosan properties before the synthesis (molecular weight, charge density/deacetylation degree)?

Line 92: ‘’ and then left for several days’’. Can the author provide more information? Could they provide an optimal value?

Line 95: The beads size and charge density/deacetylation degree should also be provided in this section. The readers might be interested to have this info ASAP in the manuscript.

Figure 1 could be rearranged into a left-right graphic to improve readability.

Varia:

The electrostatic affinities between positively charged amine groups, Cr(VI), and SDS is relatively clear in the manuscript. However, the relative proportion of Cr(VI) adsorbed directly on the chitosan backbone should be discussed deeper (e.g., on pristine chitosan beads)? I would expect some adsorption on hydroxyl groups via hydrogen bonding. Moreover, could the positively charged amino groups be neutralized by the anionic SDS character, especially under circum-neutral pH? Clear comments must be provided on existing literature:

Understanding the roles and characterizing the intrinsic properties of synthetic vs. natural polymers to improve clarification through interparticle Bridging: A review (2020)

(Table 2)

Round 2

Reviewer 1 Report

The authors have largely addressed my concerns. 

However, the English grammar of the manuscript still needs to be improved.
